# The Expression of Decidual Protein Induced by Progesterone (DEPP) Is Controlled by Three Distal Consensus Hypoxia Responsive Element (HRE) in Hypoxic Retinal Epithelial Cells

**DOI:** 10.3390/genes11010111

**Published:** 2020-01-18

**Authors:** Katrin Klee, Federica Storti, Jordi Maggi, Vyara Todorova, Duygu Karademir, Wolfgang Berger, Marijana Samardzija, Christian Grimm

**Affiliations:** 1Department of Ophthalmology, Lab for Retinal Cell Biology, University of Zurich, 8952 Schlieren, Switzerland; katrin.klee@usz.ch (K.K.); storti.federica@gmail.com (F.S.); Vyara.Todorova@usz.ch (V.T.); Duygu.Karademir@usz.ch (D.K.); Marijana.Samardzija@usz.ch (M.S.); 2Center for Integrative Human Physiology (ZIHP), University of Zurich, 8006 Zurich, Switzerland; maggi@medmolgen.uzh.ch (J.M.); berger@medmolgen.uzh.ch (W.B.); 3Institute of Medical Molecular Genetic, University of Zurich, 8952 Schlieren, Switzerland; 4Neuroscience Center, University of Zurich, 8006 Zurich, Switzerland

**Keywords:** decidual protein induced by progesterone (DEPP), hypoxia, retinal pigment epithelium (RPE), retina, hypoxia responsive element (HRE)

## Abstract

Hypoxia affects the development and/or progression of several retinopathies. Decidual protein induced by progesterone (*DEPP*) has been identified as a hypoxia-responsive gene that may be part of cellular pathways such as autophagy and connected to retinal diseases. To increase our understanding of *DEPP* regulation in the eye, we defined its expression pattern in mouse and human retina and retinal pigment epithelium (RPE). Interestingly, *DEPP* expression was increased in an age-dependent way in the central human RPE. We showed that *DEPP* was regulated by hypoxia in the mouse retina and eyecup and that this regulation was controlled by hypoxia-inducible transcription factors 1 and 2 (HIF1 and HIF2). Furthermore, we identified three hypoxia response elements (HREs) about 3.5 kb proximal to the transcriptional start site that were responsible for hypoxic induction of *DEPP* in a human RPE cell line. Comparative genomics analysis suggested that one of the three HREs resides in a highly conserved genomic region. Collectively, we defined the molecular elements controlling hypoxic induction of *DEPP* in an RPE cell line, and provided evidence for an enrichment of *DEPP* in the aged RPE of human donors. This makes *DEPP* an interesting gene to study with respect to aging and age-related retinal pathologies.

## 1. Introduction

Age-related macular degeneration (AMD) is the leading cause of irreversible visual impairment and blindness among the elderly [1,2] and its prevalence is predicted to further increase with the aging of the human population [1,3]. AMD can develop into the dry or wet form. Treatments exist only for the rarer wet form and include mainly anti-vascular endothelial growth factor (anti-VEGF) therapies [4,5,6,7] and sometimes laser photocoagulation [8]. To successfully develop new and effective therapeutics potentially targeting both forms of AMD, more insight into the mechanisms underlying the onset and progression of AMD is needed.

The retina is one the most metabolically active tissues in the human body and requires ample oxygen to generate ATP [9,10]. Age-related changes in the eye, including thickening of the Bruch’s membrane [11], reduced choroidal blood supply [12], and formation of drusen [13], may reduce the oxygen supply to the retinal pigment epithelium (RPE) and photoreceptors from the choroid [14], potentially causing a mild but chronic hypoxic state in the outer retina. Whereas short-term hypoxia can be neuroprotective against light-induced retinal degeneration [15], a chronically activated hypoxic response in photoreceptors causes retinal degeneration [16,17]. Chronic hypoxia has also been linked to the development of AMD and may be a pathogenic factor common to both wet and dry AMD [18,19].

The physiological response to hypoxia is driven by hypoxia-inducible factors (HIFs). HIF transcription factors are composed of an oxygen-regulated HIFA and a constitutively expressed HIFB subunit. Under normoxic conditions prolyl hydroxylase domain protein 2 (PHD2) hydroxylates HIFA on proline residues in the oxygen-dependent degradation domain. This causes HIFA to be recognized and ubiquitinated by a protein complex containing the von Hippel–Lindau (VHL) protein, targeting HIFA for proteasomal degradation. Under hypoxic conditions, however, HIFA is not hydroxylated, escapes degradation, and translocates to the nucleus. There, it heterodimerizes with HIFB, binds to hypoxia response elements (HREs) in the regulatory region of target genes, and drives the expression of hypoxia responsive genes together with the transcriptional co-activator p300 [20,21]. The binding of HIFs to designated HREs activates genes involved in the regulation of various metabolic pathways. Prominent examples are genes involved in erythropoiesis and angiogenesis, such as erythropoietin (*EPO*) and vascular endothelial growth factor (VEGF) (reviewed in [21]), which are upregulated in an effort to improve tissue oxygenation. Functional HREs with the consensus sequence 5′-A/GCGTG-3′ [21,22] have a preserved core of three nucleotides (5′-CGT-3′) [23,24] and can be located at positions very distant from the regulated gene. Examples are functional HREs in the EPO and PAG1 regulatory regions, located 9.2 kb and 82 kb, respectively, upstream of the transcriptional start site [25,26].

In an attempt to better understand the impact of hypoxia on the eye, we screened retinas of wild type mice exposed to acute hypoxia for differentially-regulated transcripts by genome microarrays and identified the decidual protein induced by progesterone (*Depp*) as being 35-fold upregulated in these conditions [27]. Although DEPP was first discovered as a protein induced by progesterone in endometrial stroma cells [28] and later linked to hypoxia [29], insulin signaling, and fasting [30,31,32], the physiological function of DEPP is still unknown. Recently, several publications identified DEPP as a part of the forkhead box O3 (FOXO3) – reactive oxygen species (ROS) axis that steers cell fate toward either survival or death in the neuroblastoma cell line [31,33,34]. It is therefore of interest to understand how DEPP expression is regulated to eventually modulate its level and potentially control retinal cell fate. In this study we analyzed DEPP regulation in the mouse eye, in human donor samples, and in human RPE cell lines. Using a luciferase reporter system, we identified three HREs that are responsible for the regulation of DEPP in hypoxic conditions. We also performed a comparative analysis of the identified HREs to investigate the conservation across species.

## 2. Materials and Methods

### 2.1. Human Donor Eyes

Human donor eyes were obtained from the University Eye Hospital of Zurich in accordance with the tenets of the declaration of Helsinki and after approval by the local ethics committee (BASEC-Nr: PB_2017-00550). Details of the donor eyes are given in Table 1. Peripheral nasal and central RPE were collected by carefully lifting the RPE layer from the sclera. Choroidal contamination of RPE cannot be excluded. RNA isolation and real-time PCR were performed as described below.

### 2.2. Animals

Mouse colonies were housed at the Laboratory Animal Service Center (LASC) of the University of Zurich under a light/dark cycle of 14 h/10 h with free access to food and water. Mice were euthanized with CO_2_, followed by cervical dislocation. All experiments adhered to the Association for Research in Vision and Ophthalmology (ARVO) Statement for the Use of Animals in Ophthalmic and Vision Research and were approved (Project Identification Number: ZH141/16) by the veterinary authorities of Canton Zurich, Switzerland (Kanton Zürich, Gesundheitsdirektion Veterinäramt).

### 2.3. Mouse Hypoxic Exposure and Tissue Collection

For hypoxic exposure, mice were kept for 6 h in a chamber at 7% oxygen and euthanized immediately thereafter, as previously described [35]. Control mice were kept in normal room air (normoxia) and euthanized at the same time point as the hypoxic mice to account for potential circadian variability. After euthanasia, the eyeballs were opened by a slit through the cornea using a sharp scalpel. The lens and vitreous were discarded, whereas the neural retina was extracted and immediately frozen in liquid nitrogen. The remaining eyecup (EC, containing RPE and cells of other ocular structures, such as choroid, connective tissues, and others) was separated from the eye socket using scissors and frozen in liquid nitrogen.

### 2.4. Cells

Culture conditions for Human Retinal Pigment Epithelial Cells (HRPEpiC) cells (ScienCell, Carlsbad, CA, USA) were previously described [36]. Briefly, HRPEpiC were grown on plates coated with recombinant human laminin (0.5 μg/cm^2^, BioLamina LN521, Matawan, NJ, USA). Confluent HRPEpiC were differentiated for 2–3 weeks in supplemented minimum essential media (MEM) α medium (Sigma-Aldrich, Buchs SG, Switzerland) [36]. ARPE-19 cells (ATCC, Wesel, Germany) were used between passage 24 and 30 and cultured in Dulbecco’s Modified Eagle Medium (DMEM):F12 (Gibco, LifeTechnologies, Zug, Switzerland) supplemented with 5% fetal bovine serum (FBS, Dominque Dutscher CAT S181H-500) and 10,000 U/mL penicillin–streptomycin (Gibco). ARPE-19 cells were not grown to confluency and not differentiated to allow for high transfection efficiency. All cells were kept in a humidified incubator at 37 °C with 5% CO_2_ and 21% O_2_.

U87 MG glioblastoma cells (ATCC, HTB-14) were cultured at 37 °C with 5% CO_2_ and 21% O_2_ in DMEM medium supplemented with 10% fetal bovine serum (FBS, Gibco) and 10,000 U/mL penicillin–streptomycin (Gibco).

### 2.5. In Vitro Hypoxic Exposure and DMOG Treatment of Cultured Cells

For hypoxic exposure, HRPEpiC and ARPE-19 cells were incubated in a humidified chamber at 37 °C and 0.2% O_2_/5% CO_2_ for 24 h. Normoxic control cells were kept in the same culture conditions but at 21% O_2_/5% CO_2_. Incubation for 24 h with 1 µM dimethyloxalylglycine (DMOG, diluted 1:1000 with culture medium from a 1 mM stock in Dimethyl sulfoxide (DMSO), Merck, Schaffhausen, Switzerland, D1070) was used to mimic hypoxia. Control cells were treated with matching DMSO concentrations.

### 2.6. Transfection of ARPE-19 Cells with Plasmids and siRNAs

Cells were transfected with 80 pmol specific (Table 2) or scrambled control (AllStars negative control; cat number 1027280, QIAGEN, Hilden, Germany) siRNAs and/or with 30 pmol pGL4.23[luc2/minP] plasmid (corresponding to 150 ng) (Promega, Dübendorf, Switzerland E8411) containing the DEPP regulatory region driving luciferase reporter gene expression. Co-transfection with 30 pmol of pSV-B-Galactosidase Control Vector plasmid (Promega, E1081) expressing B-Galactosidase was used to normalize for transfection efficiency. For transfection, cells were seeded in a 24-well plate at a density of 10,000 cells/well in 1 mL of cell culture medium. Transfections were done in 950 µL of cell culture medium supplemented with 50 µL of an Opti-MEM (Gibco)/Lipofectamine (Invitrogen, Thermo Fischer, Reinach, Switzerland) mix containing nucleic acids as described above for a duration of 24 h.

Plasmid transfection was performed with the Lipofectamine 3000 kit (Invitrogen, Thermo Fischer), whereas siRNA transfection was done using the Lipofectamine RNAi MAX kit (Invitrogen, Thermo Fischer). Briefly, ARPE-19 cells were passaged one day after seeding and transfected with 30 pmol of plasmid DNA or 80 pmol of siRNA per well. After 24 h of incubation, the transfection medium was replaced by fresh medium and cells were incubated for another 24 h. Thereafter, cells were either exposed to hypoxia (0.2% O_2_), kept in normoxia. or treated with DMOG 1 µM for 24 h.

For combined siRNA and plasmid transfections, ARPE-19 cells were first transfected with 80 pmol of siRNA using the Lipofectamine RNAi MAX kit (Invitrogen, Thermo Fischer). After 6 h, the transfection medium was replaced with fresh medium and cells were transfected with 30 pmol of plasmid using the Lipofectamine 3000 kit (Invitrogen, Thermo Fischer). After 24 h, the transfection medium was replaced by normal cell culture medium and cells were incubated for another 24 h before they were exposed to hypoxia, kept at normoxia, or treated with DMOG for 24 h.

### 2.7. RNA Isolation, cDNA Synthesis, and Real-Time PCR

RNA from mouse and cell culture samples was isolated using the NulceoSpin RNA kit (RNA isolation kit Macherey-Nagel, Oensingen, Switzerland), while RNA from human donor samples was purified with the RNeasy Mini Kit (QIAGEN) following the manufacturer’s instructions. Complementary DNA (cDNA) synthesis from total RNA was done using an oligo dT primer and murine leukemia virus (MLV) reverse transcriptase (Promega). A total of 10 ng of cDNA was used for real-time PCR using an ABI QuantStudio3 device (Thermo Fisher Scientific) and the PowerUp SYBR Green 449 master mix (Thermo Fisher Scientific). Primer pairs are listed in Table 3. Transcript levels were calculated using *ACTB* (human) or *Actb* (mouse) as internal reference genes and the ΔΔCt method provided by the Thermo Fischer Cloud.

### 2.8. Western Blotting

After washing with phosphate-buffered saline (PBS), cells were collected into 100 µL of Lämmli sample buffer, incubated at 65 °C for 10 min, and centrifuged at full speed for 5 min. Supernatants were collected and stored at −20 °C until further use. Western blotting was done as described recently [16]. Briefly, samples were thawed at 37 °C and proteins in 30 µL of the lysates were separated by a 12% sodium dodecyl sulfate polyacrylamide gel electrophoresis (SDS-PAGE). Separated proteins were transferred to a nitrocellulose membrane using a semi-dry blotting system (BioRad, Reinach, Swtizerland). Membranes were blocked with 5% non-fat dry milk (BioRad) for 1 h at room temperature and incubated overnight with primary antibodies (Table 4) diluted in blocking milk at 4 °C. Conjugated secondary antibodies (Table 4) were applied for one hour in blocking milk at room temperature. Enhanced chemiluminescence (ECL) substrate (PerkinElmer, Schwerzenbach, Switzerland) was used to develop the signals, which were visualized on X-ray films.

### 2.9. Luciferase/β Galactosidase Assay

Cells were lysed with 100 µL of Passive Lysis Buffer (PLB, Promega), and transferred to a 1.5 mL centrifuge tube. Lysates were vortexed for 15 s and centrifuged for 30 s at 12,000× *g*. Supernatants were collected and frozen at −80 °C until further use. For the luciferase assay, 10 µL of lysates were pipetted into a black 96-well plate. A total of 50 µL of luciferin master mix (Luciferase Assay System, Promega) was added and luminesce was immediately measured with a plate reader (BioTek, Synergy HT, Sursee, Switzerland). β galactosidase colorimetric reaction was used as transfection control. A total of 50 µL of ortho-nitrophenyl-β-galactoside (ONPG, Thermo Scienfitic) solution (3 mM in phosphate buffer supplemented with 0.1 mM β-mercaptoethanol and 10 mM MgCl2) was added to 10 µL of lysates in a transparent 96-well plate and incubated for 60 min at 30 °C. Absorbance was measured at 405 nm using a plate reader (BioTek, Synergy HT). All samples were run in duplicates. Luminescence counts (luciferase activity) of the duplicates were averaged and divided by the optical density (OD) reads (b-galactosidase activity), yielding normalized values, which are presented as fold-changes in the graphs.

### 2.10. In-Silico Identification of Putative Hypoxia Responsive Elements (HRE)

To identify putative HREs within the 3.8 kb upstream region of the human DEPP transcription start (TSS) site, we used the JASPAR database (http://jaspar.genereg.net/). The JASPAR algorithm identified 29 putative consensus for Aryl Hydrocarbon Receptor Nuclear Translocator (ARNT):HIF (Appendix A).

### 2.11. Cloning of the 3.8 kb Region Upstream of DEPP Transcriptional Start Site, Deletion Constructs and Sit- Directed Mutagenesis

Human genomic DNA (gDNA) isolated from ARPE-19 cells was used as a template to amplify the 3.8 kb region upstream of DEPP transcription start site (TSS) with Phusion High-Fidelity Polymerase (New England Biolabs, Allschwill, Switzerland, M0530S) using the primer pair listed in Appendix A. Two restriction sites (KpnI, NheI; underlined sequence in primers listed in Appendix A) and extra nucleotides (indicated by small letters in primers listed in Appendix A) were added to clone the fragment into pGL4.23[luc2/minP] (Promega, E8411) using T4 ligase (New England Biolabs, M0202S). The sequence (Microsynth, Balgach, Switzerland) of the cloned fragment is given in Appendix A. The resulting pGL4.23-hDEPP 3.8 kb plasmid served as the source for all deletion constructs and site-directed mutagenesis. Phusion High-Fidelity Polymerase (New England Biolabs, M0530S), the primers listed in Appendix A, and cloning of the amplified fragments into the KpnI and NheI restriction sites were used to generate the plasmids used in this study. The BLOCK1 + 3 plasmid was constructed in two steps. First, BLOCK1 and 3 were amplified separately. BLOCK1 was flanked with SacI and NheI restriction consensus and BLOCK 3 was flanked by KpnI and SacI restriction consensus. Then, BLOCK1 was cloned into an empty pGL4.23[luc2/minP] vector (Promega, E8411), followed by cloning BLOCK3 upstream of BLOCK1.

The New England Biolabs (NEB) Q5 Site-Directed mutagenesis kit (New England Biolabs), was used to mutate the core consensus of the selected HREs, precisely following the manufacturer’s instructions. Primers used to mutate the HREs are listed in Appendix A. All plasmids were verified by sequencing (Microsynth).

### 2.12. Conservation Analysis

Conservation analysis was performed on the 3.8 kb region upstream of DEPP exon 1 (chr10:45474246-45478004 on the hg19 build) based on phastCons100way scores [37], which were downloaded from the UCSC Table Browser [38]. The 154 bp region, including the putative HREs 25, 26, and 27 (chr10:45477663-45477816 on the hg19 build) was aligned to the homologous regions of rhesus macaque (Macaca mulatta, chr9:43217687-43217842 on the rheMac3 build), mouse (Mus musculus, chr6:116647851-116648019 on the mm10 build), and pig (*Sus scrofa*, chr14:99578454-99578601 on the susScr3 build) using the MultAlin software [39].

### 2.13. Statistical Analysis

All statistical analyses were done using PRISM software (Graphpad Software, San Diego, CA, USA). Tests used to test significance for individual results are given in the figure legends.

## 3. Results

### 3.1. DEPP Is Upregulated in Hypoxic Ocular Tissues and Cells

We recently identified Depp as the second-top regulated gene in the hypoxic mouse retina using a gene chip assay [27]. To validate these data, we tested *Depp* expression by real-time PCR in normoxic and hypoxic retinas and eyecups (EC, containing the RPE) from wild type mice. *Depp* was similarly upregulated in both mouse tissues that were exposed to acute hypoxia (Figure 1a, left panel). This upregulation was comparable to the upregulation of adrenomedullin (*ADM*), a known HIF target gene (Figure 1a, right panel). To establish an in vitro model for the investigation of Depp regulation by hypoxia, we determined its expression in human primary RPE cells HRPEpiC and in a human RPE cell line (ARPE-19) (Figure 1b). Both cell systems increased the expression of *DEPP* under hypoxic culture conditions (0.2% O_2_), with strong induction after 24 h (Figure 1b, left panels). This pattern resembled the expression of *ADM* and vascular endothelial growth factor (*VEGF*), two known HIF-target genes (Figure 1b, right panels). The strong expression of *DEPP* and *ADM* in HRPEipC cells in the presence of the prolyl hydroxylase (PHD) inhibitor and hypoxia mimetic dimethyloxalylglycine (DMOG) suggests that upregulation depends on the PHD–HIF pathway (Figure 1b, top panels).

Considering the potential development of hypoxia in the aging eye, we tested the expression of *DEPP* in RPE tissue collected from the eyes of human donors aged between 17 and 90 years (Figure 1c, Table 1). Although the expression of *DEPP* in the peripheral RPE was not different between young and old donors (linear regression, *p* = 0.0875), *DEPP* levels in the central human RPE showed a significant enrichment in older subjects (linear regression, *p*-value = 0.0175) suggesting that *DEPP* may be differentially regulated in the aged central human RPE, potentially in response to reduced tissue oxygenation or other age-dependent alterations. It is worth noticing that the donors who suffered from brain trauma or cerebral ischemia (Table 1) did not show particularly high *DEPP* levels (not shown), indicating that the observed *DEPP* upregulation in the aged RPE may not be due to an acute ischemic pathology.

### 3.2. DEPP Is a HIF1 and HIF2 Target Gene in the ARPE-19 Human Cell Line

The upregulation of *DEPP* levels by hypoxia and DMOG (Figure 1) indicates that *DEPP* expression might be controlled by HIF transcription factors. To test this, we used siRNAs to knockdown *HIF1A*, *HIF2A*, or both simultaneously in normoxic and hypoxic ARPE-19 cells. Both siRNAs showed efficient downregulation of their respective target RNA (Figure 2a) and proteins (Figure 2b), both in normoxia and hypoxia. Although knockdown of either *HIF1A* or *HIF2A* alone did not abolish the hypoxia-mediated induction of *DEPP* expression, the simultaneous knockdown of both transcription factors completely prevented *DEPP* upregulation under hypoxic conditions (Figure 2c). The same *HIF* dependency was detected for the expression of the hypoxia-responsive *ADM* gene (Figure 2c).

### 3.3. A 3.8 kb Fragment Upstream of the Transcription Start Site Is Sufficient for Hypoxic Regulation of DEPP Expression

The 1000 bp DNA fragment immediately upstream of the human *DEPP* transcriptional start site (TSS) does not confer hypoxic regulation of *DEPP* [40]. We cloned a larger, 3.8 kb DNA fragment of the human *DEPP* promoter into a luciferase reporter vector (pGL4.23) to study the response of this larger regulatory region to hypoxia. ARPE-19 cells transfected with the reporter vector carrying this 3.8 kb fragment (pGL4.23-hDEPP 3.8 kb) displayed 4- to 5-fold increased luciferase activity under hypoxic conditions when compared to normoxic controls (Figure 3). Similar to the regulation of *DEPP* expression in ARPE-19 cells (Figure 2c), hypoxic induction of luciferase activity was completely abolished in the presence of siRNAs against both *HIF1A* and *HIF2A* (pGL4.23-hDEPP 3.8 kb siHIFs), validating the reporter system used in this study (Figure 3a). The inspection of the 3.8 kb regulatory region for the presence of HREs using the online JASPAR database identified 29 putative HREs (Figure 3b, Appendix A). To identify functional HREs responsible for the hypoxic induction of *DEPP*, the 3.8 kb fragment was split into three blocks (see Figure 3b for representation) and cloned in different combinations into the reporter vector (Figure 3c, Appendix A). The presence of BLOCK3, either alone or in combination with BLOCK1 or BLOCK2, was essential for the hypoxic induction of luciferase activity in transfected ARPE-19 cells (Figure 3c). This suggests that critical HRE elements are located in BLOCK3 of the *DEPP* regulatory sequence. To further narrow the sequences responsible for hypoxic induction, we truncated the 3.8 kb fragment, starting from the 5′ end where BLOCK3 resides. The constructs containing HREs 1–19 or 1–24 were unresponsive to hypoxia (Figure 3d). In other words, the removal of six distal HREs from the 3.8 kb regulatory sequence of DEPP was sufficient to abolish the response to hypoxia. Hypoxic induction was restored upon the addition of HRE 25 (Figure 3d), albeit of a lower magnitude as compared to the full length 3.8 kb fragment containing all 29 HREs (Figure 3d). We therefore concluded that the distal region containing HREs 29–25 is likely involved in the hypoxic regulation of *DEPP*.

### 3.4. Disruption of HREs 25, 26, and 27 Individually or in Combination Abolishes the Hypoxic Response

To investigate the contribution of HREs 25–29 to hypoxic induction of *DEPP*, we mutated each of these HREs individually in the full-length construct. The HRE consensus sequence consists of five bases with a preserved core of three nucleotides (5′-CGT-3′) that confers HIF binding. We mutated the core consensus into 5′-TAC-3′, as this particular sequence was shown to completely abolish HIF-binding [23,24]. Mutagenesis of HRE 20 was used as control, since it should not affect the hypoxic response of the 3.8 kb regulatory region (Figure 3d).

The mutation of HREs 20, 28, or 29 did not change hypoxic induction of luciferase activity. Mutating HREs 25, 26, or 27, however, completely abolished the hypoxic induction of luciferase activity (Figure 4a), suggesting that each of these three HREs (Figure 4b) is critical for hypoxic upregulation of *DEPP*. Furthermore, a similar effect was observed when DMOG was used instead of hypoxia to induce the expression of luciferase reporter. The simultaneous mutation of HREs 25, 26, and 27 completely prevented the induction of luciferase activity also by DMOG treatment (Figure 4c). Collectively, our data suggest that it is likely that HREs 25–27 drive the hypoxic response by a mechanism that directly or indirectly involves HIF transcription factors.

### 3.5. HRE 25 Is Conserved across Species

We then investigated if HREs 25 to 27 responsible for activity of the DEPP regulatory region in hypoxic conditions are evolutionary conserved. The whole 3.8 kb fragment from human was plotted against the phastCons100way scores. Regions with high conservation scores did not overlap with putative HREs, except for the genomic portion encompassing HRE 25 (Figure 5a). Therefore, we aligned 154 bp, encompassing HRE2 27, 26, and 25, against the corresponding genome regions of macaque (Rhesus Macaque), mouse (Mus Musculus), and pig (Sus scrofa) and plotted with the corresponding conservation score (phastCons100way). A detailed analysis of this genomic region revealed a high evolutionary conservation of up to 17 nucleotides, including HRE 25 and its flanking region (Figure 5b). Although mutating HREs 26 and 27 abolished hypoxic induction in human cells similar to mutating HRE 25 (Figure 4a), HREs 26 and 27 did not localize to a conserved region (Figure 5b).

## 4. Discussion

Considering the potential link between reduced tissue oxygenation and various retinopathies, it is important to study hypoxia-driven mechanisms to understand retinal pathologies as a prerequisite to develop potential therapies. In this work, we showed that the hypoxia-responsive gene *DEPP* [29] is expressed in the neural retina and RPE, identified the regulatory elements driving its hypoxic induction, and showed that both HIF1 and HIF2 transcription factors are responsible for its hypoxic regulation.

Although not much is known about the function of *DEPP*, it is an interesting gene to study, since it can be regulated by many different stimuli, including progesterone, hypoxia, nutrient deprivation, retinoid X receptors (RXR) agonists, and insulin [28,29,30,32,34,40,41,42]. The *DEPP* promoter, previously defined in a human neuroblastoma cell line as the −1116 bp sequence upstream of *DEPP* TSS, was shown to respond to FOXO3 transcription factors. Three functional FOXO3 binding sites were identified and each of them contributes to *DEPP* expression [31]. Although the hypoxic regulation of *DEPP* has been recognized before [29], the responsible regulatory elements had not been identified. A recent study merely showed that the 1000 bp DNA fragment that contains the three FOXO3 binding sites does not confer hypoxia-inducibility [40]. In line with those data, we did not observe hypoxic activation by the BLOCK1 and 2 sequences, which encompass a total of 2783 bp upstream of the *DEPP* TSS. The absence of functional HREs in close proximity to the TSS of a hypoxia-responsive gene is quite common, as whole-genome studies have shown that more than 60% of HIF-target genes display functional HREs located more than 2.5 kb upstream of the TSS [26,43]. Consistently, we identified three HREs (25, 26 and 27) located between −3424 and −3558 bp upstream of the TSS to be relevant for hypoxic induction of *DEPP*. Each of these HREs was necessary, as site-directed mutagenesis of either one alone abrogated the response to hypoxia. However, chromatin immunoprecipitation (ChIP) experiments will be performed to prove binding of the HIF transcription factors to the identified *DEPP* HREs.

Since HIF-regulated genes can possess more than one functional HRE, and each HRE can contribute to a different extent to HIF-driven gene expression (reviewed in [44]), it was surprising that the individual mutagenesis of HRE 25, 26, and 27 completely abolished the hypoxic induction of the reporter gene (Figure 4a). It has been reported that closely spaced HREs, for example, in the regulatory region of the human 6-phosphofructo-2-kinase/fructose-2,6-bisphosphatase (*HP2K*) gene lose their ability to induce transcription when mutated individually, suggesting that these HREs may act as a unit [45]. Although the spacing of the HREs in *HP2K* was only four nucleotides, these data indicate that individual HREs may collaborate to form functional entities. Whether this is also the case for DEPP HREs 25, 26, and 27, which are spaced by 84 bp and 26 bp (Appendix A), still needs to be determined, but it is likely that specific DNA folding and/or DNA-protein complexes in the region of HREs 25–27 are part of the regulation of DEPP expression in hypoxia. Interestingly, comparative analysis on the genomic area encompassing HREs 25–27 revealed that HRE25 was evolutionarily conserved among humans, macaque, mouse, and pig, whereas HREs 26 and 27 were not. This may indicate that HRE 25 is necessary and sufficient to drive hypoxic upregulation in those species, a hypothesis that needs to be addressed in further studies. We also identified a highly evolutionary conserved sequence between HRE 25 and HRE 26 that contains an NF–E2 binding site (5′-GCTGAGTCA-3′) [46]. Interestingly, binding of the transcription factor nuclear factor, erythroid 2 like-2 (NRF2) to the NF–E2 consensus sequence was shown to drive the expression of antioxidant genes under hypoxic conditions [47].

Although lack of *DEPP* did not affect general mouse development and aging [48], potential effects on retinal morphology and function after post-natal day five have not been specifically evaluated in *DEPP* knockouts. This leaves the question of a possible involvement of *DEPP* in retinal pathologies and aging still open. It is interesting to note that the expression of *DEPP* increased in the central human RPE with age (Figure 1c) and that a recent report proposed that increased *DEPP* levels lead to the inhibition of catalase activity, resulting in ROS accumulation and the activation of autophagy [31,33]. Since impaired autophagy in the RPE can lead to macrophage recruitment, activation of inflammasomes, degeneration of RPE and photoreceptors, and to the development of AMD [49], increased expression of *DEPP* might contribute to mechanisms that support the survival of RPE cells in the aging eye. Although this is speculative and warrants further experimentation, it may be worthwhile exploring the function and regulation of DEPP in detail with respect to retinal pathologies.

In summary, we characterized *DEPP* as an HIF-regulated gene in the retina and RPE. Analysis of a 3.8 kb DNA fragment 5′ of the human *DEPP* TSS identified three functional HREs located between −3424 and −3558 bp upstream of the TSS. Each of the three HREs was required for the hypoxic activation of a downstream coding sequence by the 3.8 kb regulatory region, suggesting that they may influence each other and act in concert. The specific regulation of *DEPP*, its increased expression in the central RPE with age, and its connection to oxidative stress and autophagy asks for studies that investigate a potential contribution of *DEPP* to retinal pathologies, including AMD.

## Figures and Tables

**Figure 1 genes-11-00111-f001:**
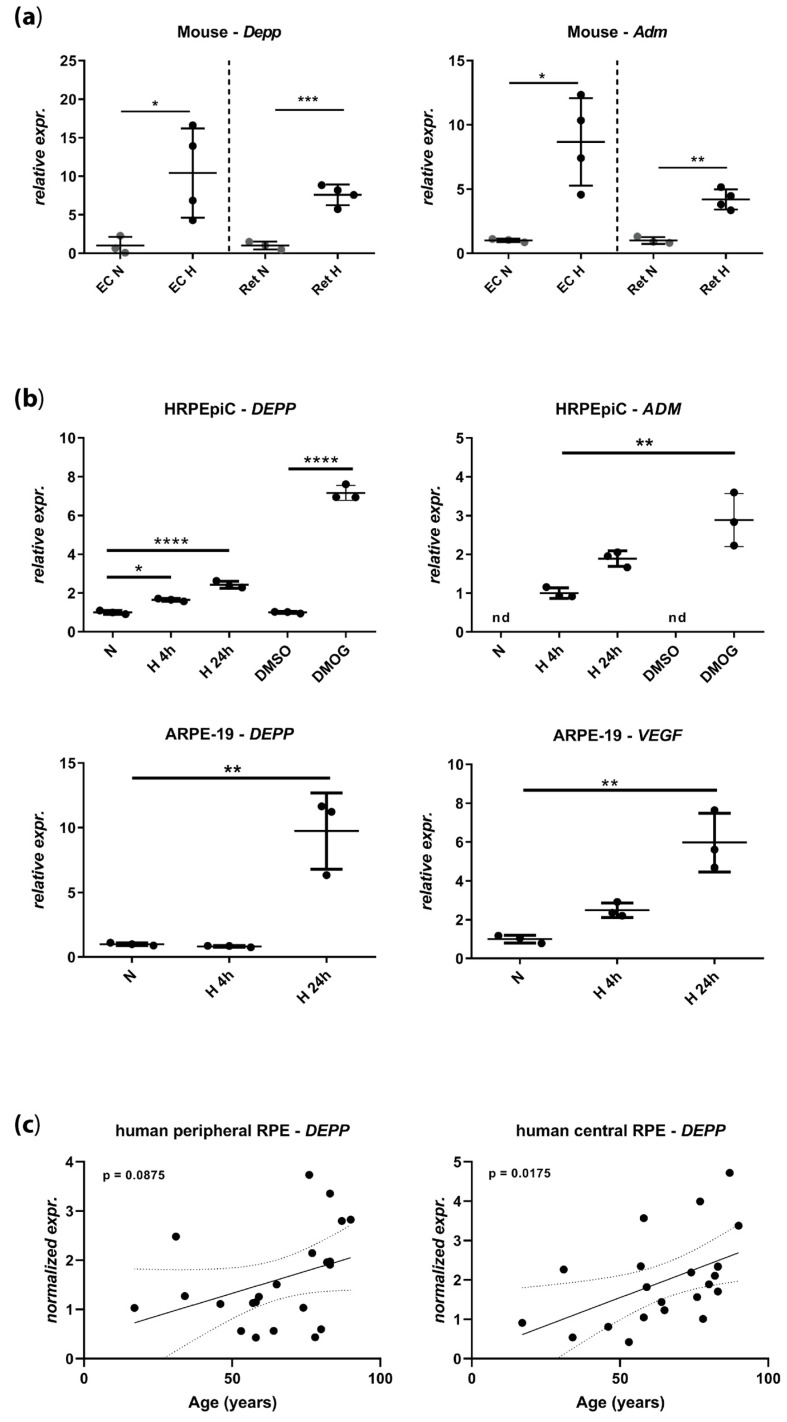
Expression levels of decidual protein induced by progesterone (*DEPP*) and known hypoxia-inducible factors (HIF) target genes by semi-quantitative real-time PCR in (**a**) mouse eyecups (EC) and retina (normoxia: *N* = 3 eyes of three mice; hypoxia: *N* = 4 eyes of three mice) and (**b**) human retinal pigment epithelial cells (HRPEpiC) and human RPE cell line ARPE-19 cells. Levels in cells exposed to hypoxia (H) and cells treated with dimethyloxalylglycine (DMOG) are shown as fold changes over normoxic (*N*) or DMSO controls, respectively. Note that the transcript levels of adrenomedullin (*ADM*) in HRPEpiC are expressed relatively to the levels of 4 h of hypoxia. (**c**) *DEPP* transcript levels determined by semi-quantitative real-time PCR in human retinal pigment epithelium (RPE) samples collected from central and peripheral areas of donor eyes. Donor age ranged from 17 to 90 years (Table 1; *N* = 22). EC: Eyecup, Ret: Retina, *Adm*: Adrenomedullin, *VEGF*: Vascular endothelial growth factor, *N*: Normoxia, H: Hypoxia, nd: Not detected. Shown are individual values and means ± SD, Statistics: (A) and (B) one-way ANOVA with the Brown–Forsythe test, (C) linear regression. Dotted line indicates the *p*-value field. *: *p* ≤ 0.05, **: *p* ≤ 0.005, ***: *p* ≤ 0.005, ****: *p* ≤ 0.0005.

**Figure 2 genes-11-00111-f002:**
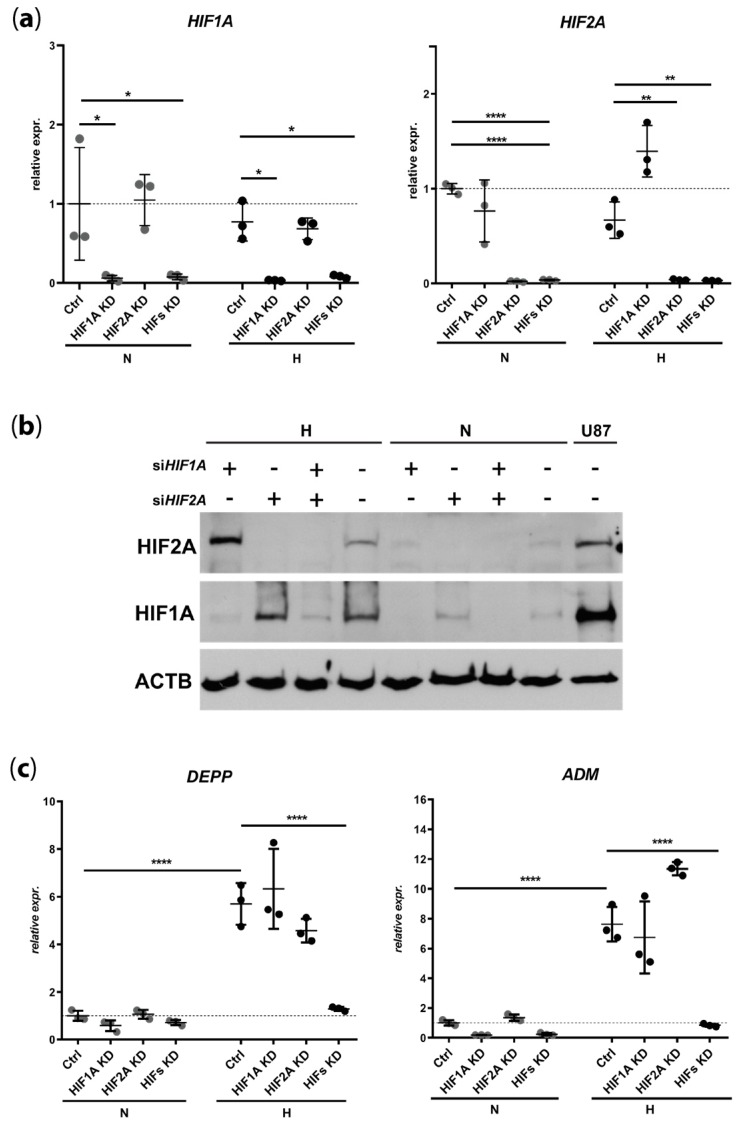
Loss of *DEPP* induction upon *HIF1A* and *HIF2A* knockdown in ARPE-19 cells. *HIF1A* and *HIF2A* (**a**), *DEPP* and *ADM* (**c**), transcript levels in normoxia (*N*) and hypoxia (H) were determined by real-time PCR in presence of *HIF1A* siRNA (HIF1A KD), *HIF2A* siRNA (HIF2A KD), or both *HIF1A* and *HIF2A* siRNAs (HIFs KD). Data are shown as fold changes over normoxic (N) siCTRL (Ctrl). (**b**) HIF1A and HIF2A protein levels in cells treated with siRNAs as indicated. Beta Actin (ACTB) served as loading control. Normoxic U87 neuroblastoma cells were used as positive control for HIF1A and HIF2A. Abbreviations see Figure 1. Shown are individual values and means ± SD of *N* = 3. Statistics: One-way ANOVA with Brown–Forsythe test, *: *p* ≤ 0.05, **: *p* ≤ 0.005, ***: *p* ≤ 0.005, ****: *p* ≤ 0.0005.

**Figure 3 genes-11-00111-f003:**
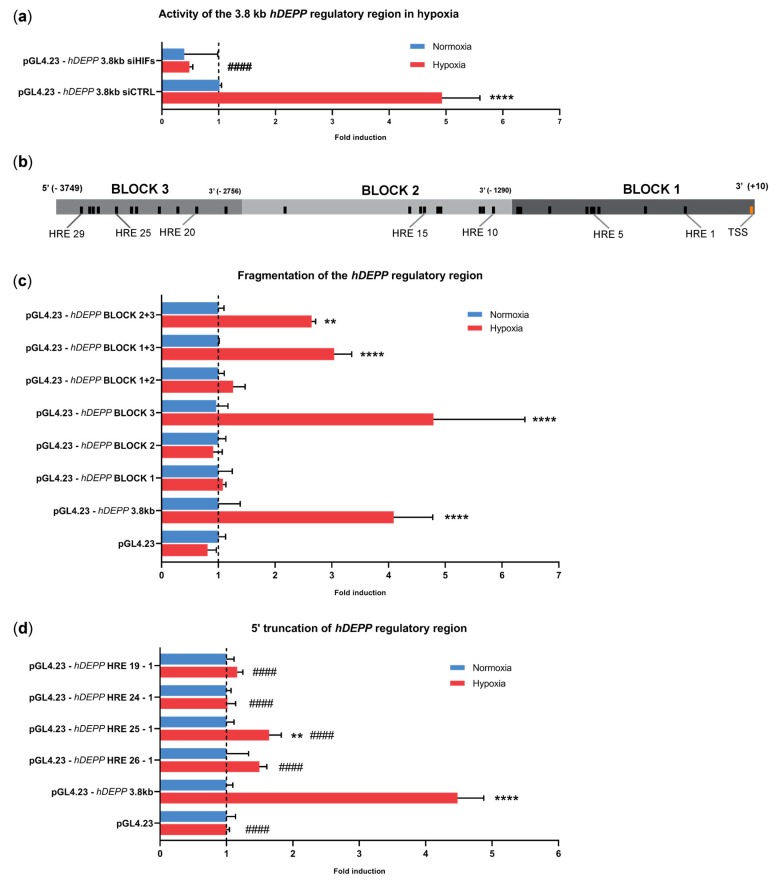
Hypoxic regulation by the 3.8 kb DNA fragment upstream of the *DEPP* transcriptional start site. (**a**) Luciferase activity driven by the 3.8 kb h*DEPP* regulatory region assayed in hypoxic and normoxic ARPE-19 cells. Cells were transfected with the reporter plasmid containing *DEPP* regulatory sequence and treated with scrambled siRNA (siCTRL) or with siRNAs against *HIF1A* and *HIF2A* (siHIFs). Luciferase activity was expressed relatively to the activity in normoxic ARPE-19 cells treated with control siRNA (pGL4.23-*hDEPP* 3.8 kb siCTRL). (**b**) Schematic representation of the 3.8 kb DNA fragment isolated from ARPE-19 genomic DNA and cloned into the luciferase reporter vector pGL4.23. Putative HREs are shown as black bars. The HRE closest to the TSS is numbered as HRE 1 and the most distal one as HRE 29. Precise positions of the HREs are given in Appendix A. The DNA fragment was subdivided into three BLOCKS, as indicated. (**c**) Luciferase activity in cells transfected with the indicated BLOCK constructs was expressed relative to the respective normoxic controls, which were set to 1. (**d**) Luciferase activity in cells transfected with 5′ truncated constructs. HRE numbers in the construct name indicate the presence of the respective HRE sequences. Luciferase activities were expressed relative to the respective normoxic controls, which were set to 1. All luciferase values were normalized to b-galactosidase activity from the co-transfected normalizer plasmid. The empty pGL4.23 plasmid served as negative control. HRE: hypoxia responsive element. Shown are means ± SD of *N* = 3. Statistics: One-way ANOVA with the Brown–Forsythe test, *: *p* ≤ 0.05, **: *p* ≤ 0.005, ***: *p* ≤ 0.005, ****: *p* ≤ 0.0005, ####: *p* ≤ 0.0005. Stars (*) denote significance between the hypoxic and normoxic activity of the indicated vector. Hashtags (#) indicate significance between the hypoxic activity of the indicated vector and hypoxic activity of the reporter vector containing the 3.8 kb regulatory region (pGL4.23-hDEPP3.8 kb).

**Figure 4 genes-11-00111-f004:**
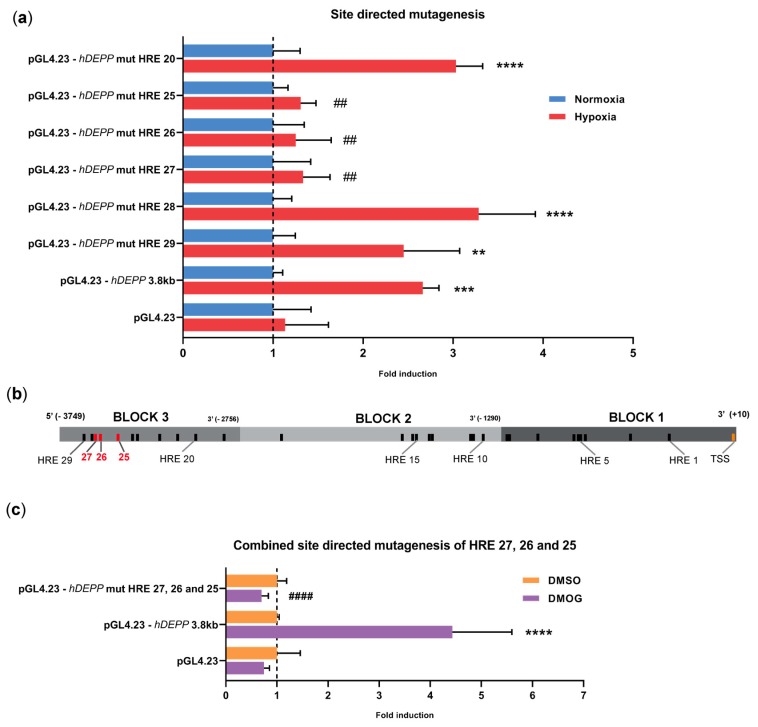
Identification of three functional HREs for *DEPP* hypoxic regulation. (**a**) Luciferase activity of cells transfected with the reporter plasmid containing the *wt* or the mutated 3.8 kb *DEPP* regulatory region. The mutated HREs are indicated for each construct. Normoxic activity of the constructs was set to 1 and hypoxic activity was expressed relative to the activity in normoxia. (**b**) Schematic representation of the relevant HREs (red) for hypoxic induction in ARPE-19 cells. (**c**) Luciferase activity conferred by the 3.8 kb *DEPP* fragment carrying the three mutated HREs 25, 26, and 27 after treatment with DMOG. All luciferase values were normalized to b-galactosidase activity from the co-transfected normalizer plasmid. The empty pGL4.23 plasmid served as negative control. Shown are means ± SD of *N* = 3. One-way ANOVA with the Brown–Forsythe test, * or #: *p* ≤ 0.05, ** or ##: *p* ≤ 0.005, *** or ###: *p* ≤ 0.005, **** or ####: *p* ≤ 0.0005. Stars (*) refer to the comparison of luciferase activity in hypoxic and normoxic cells. Hashtags (#) indicate significance between hypoxic activity of the indicated vector and hypoxic activity of the reporter vector containing the 3.8 kb regulatory region (pGL4.23-hDEPP3.8 kb).

**Figure 5 genes-11-00111-f005:**
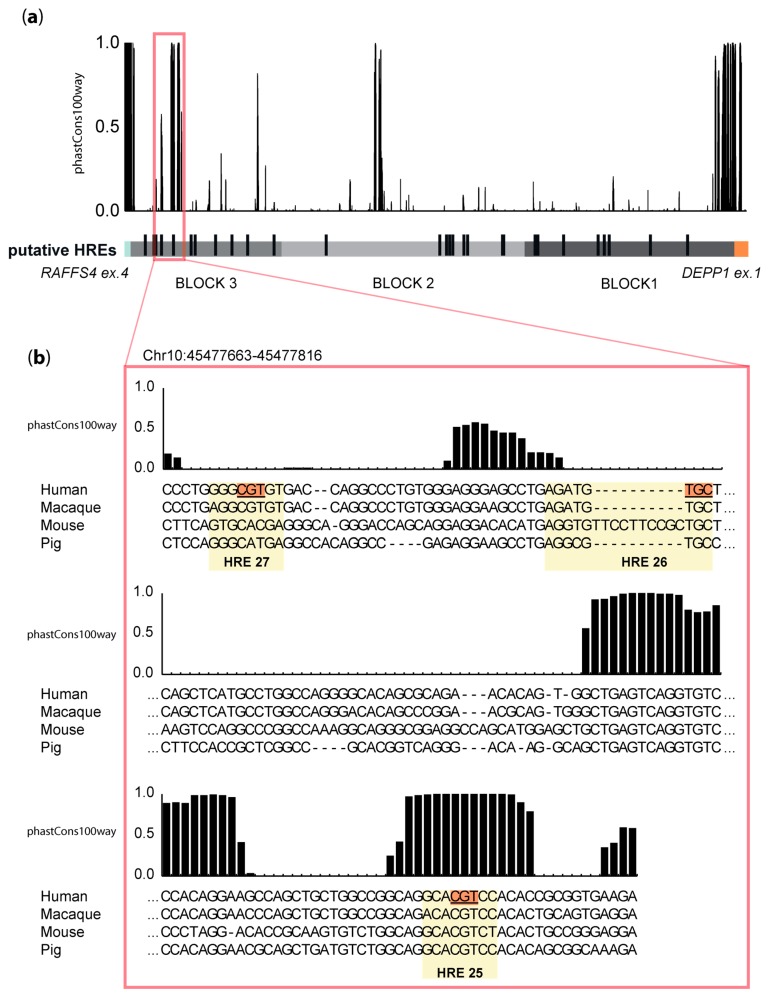
Analysis of DNA sequence conservation upstream of *DEPP* TSS. (**a**) Schematic representation of the 3.8 kb region analyzed (chr10:45474246-45478004 on the hg19 build), divided in three blocks (bottom). The putative HREs are indicated by black lines, whilst the three blocks are shown in different tones of grey. The plot on top of the region shows the corresponding conservation score (phastCons100way). A value of 1.0 indicates 100% conservation across the species tested. (**b**) Detailed view of the region containing HREs 25–27, which were identified as being responsible for the hypoxic induction *DEPP* transcription (chr10:45477663-45477816 on the hg19 build). The human sequence (hg19) was aligned to the homologous region of rhesus macaque (Macaca mulatta, rheMac3), mouse (Mus musculus, mm10 build), and pig (*Sus scrofa*, susScr3). The putative HREs identified in this region are highlighted in light yellow and their core sequence underlined and highlighted in orange. On top of each nucleotide, the plot displays the corresponding conservation score (phastCons100way).

**Table 1 genes-11-00111-t001:** Human donor specifications.

Age	Gender	Known Disease	Cause of Death
17	M	-	Brain trauma and multiple contusions
31	M	-	Trauma
34	F	Alcohol addiction	Brain trauma and multiple contusions
46	M	Leukaemia	Chickenpox (varicella) virus (VZV) infection with encephalitis
53	M	Hypertension	Brain hemorrhage
57	M	-	Hemorrhage
58	M	Colitis ulcerosa	Brain trauma
58	F	Depression	Acute liver failure
59	F	Diabetes Type 1	Heart failure
64	M	-	Heart failure
65	M	Diabetes Type 2	Hemorrhage
74	M	Adenocarcinoma	Prostate adenocarcinoma
76	M	Adenocarcinoma	Adenocarcinoma at the intestine + lung metastasis
77	M	-	Brain hemorrhage
78	M	Coronary disease	Heart failure
80	F	Coronary disease	Cerebral ischemia
82	M	Adenocarcinoma	Gastroesophageal adenocarcinoma
83	F	-	Septic shock
83	F	Hypertension	Cerebral ischemia
83	F	Epilepsy	Brain trauma
87	M	Pancreas carcinoma	Collapse
90	M	-	Heart failure

**Table 2 genes-11-00111-t002:** siRNA sequences.

siRNA	Sequence (5′-3′)	Antisense (5′-3′)
siHIF1A	CUGAUGACCAGCAACUUGA-dTdT	UCAAGUUGCUGGUCAUCAG -dTdT
siHIF2A	CAGGUGGAGCUAACAGGACAUAGUA-dTdT	UACUAUGUCCUGUUAGCUCCACCUG -dTdT

**Table 3 genes-11-00111-t003:** semi quantitative RT-PCR primer.

Gene	Forward (5′-3′)	Reverse (5′-3′)	Product Length
*Adm (mouse)*	TCCTGGTTTCTCGGCTTCTC	ATTCTGTGGCGATGCTCTGA	133 bp
*ADM (human)*	TTGGACTTCGGAGTTTTGCC	CCCACTTATTCCACTTCTTTCG	149 bp
*Actb (mouse)*	CAACGGCTCCGGCATGTGC	CTCTTGCTCTGGGCCTCG	153 bp
*ACTB (human)*	CCTGGCACCCAGCACAAT	GGGCCGGACTCGTCATAC	144 bp
*Depp (mouse)*	CCCTGACTGCTGACTTACA	TTCCCGAATCGTTGGCA	76 bp
*DEPP (human)*	TGTCCCTGCTCATCCATTCTC	CTCACGTAGTCATCCAGGC	199 bp
*HIF1A (human)*	TTCACCTGAGCCTAATAGTCCC	TCATCCATTGATTGCCCCAGCAGTC	276 bp
*HIF2A (human)*	CGGAGAGGAGGAAGGAGAAG	AGAGCAAACTGAGGAGAGGAG	191 bp
*VEGF (human)*	GTGGACATCTTCCAGGAGTACC	TGTTGTGCTGTAGGAAGCTCAT	205 bp

**Table 4 genes-11-00111-t004:** Antibodies.

Antibody	Dilution	Catalogue Number	Company
Anti-HIF1A	1:2000	NB100-479	Novus Biologicals
Anti-HIF2A	1:2000	PAB12124	Abnova
Anti-ACTB	1:10,000	A5441	Sigma-Aldrich
HRP conj. Sec. Antibody anti-Mouse	1:10,000	sc-2031	Santa-Cruz
HRP conj. Sec. Antibody anti-Rabbit	1:10,000	#7074	Cell Signalling

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
