# Peer review of "The Expression of Decidual Protein Induced by Progesterone (DEPP) Is Controlled by Three Distal Consensus Hypoxia Responsive Element (HRE) in Hypoxic Retinal Epithelial Cells"

_genes, 2020, doi:10.3390/genes11010111_

Round 1

Reviewer 1 Report

Klee and coworkers report on hypoxia-driven expression of DEPP in human and mouse ocular tissues and RPE cell lines along with the identification of promotor sequences driving the response of the gene to hypoxia.

The deciphering of hypoxia-driven mechanisms in the eye certainly deserves consideration both to advance knowledge of eye functioning and therapeutic perspectives. The paper is very well written and easy to follow. It reports a beautiful piece of work which includes a large variety of techniques, making the results and conclusions rather encompassing.

I have only some minor points of concern.

1. Analysis of DEPP abundance in human eye donors

-“DEPP levels in the central human RPE showed a significant enrichment in older subjects (linear regression, p value = 0.0175) suggesting that DEPP may be differentially regulated in the aged central human RPE, potentially in response to reduced tissue oxygenation or other age-dependent alterations.”

Several of the oldest individuals succumbed Brain trauma or cerebral ischemia. Whether this had a negative impact on eye perfusion before patient death needs to be taken into account. It would be useful to include RTqPCR of markers of acute retina ischemia. As it would be interesting to know the abundance of ADM in the samples.

-The gene used to normalize the data is not mentioned. I guess this is beta actin but the reference should be mentioned here and wherever necessary.

2. Analysis of mouse RPE.

Although the p value is ≤ 0.05 when comparing DEPP expression in eye cups from mice maintained in hypoxia versus normoxia there is a great variability. It is unclear whether the 4 points correspond to 8 eyes of 4 mice, 4 eyes of 4 mice or 4 eyes of 2 mice.

3. HIF1A knock-down in ARPE-19 cell lines. Again, while p values are ≤ 0.05 there exist a high dispersion of the data, the number of experiments could have been increased to 5 to improve significance

4. The results of the DEPP promotor analysis using a tradition reporter assay-based design are supportive of the conclusions. However, it would have been nice to introduce mutations in of the most relevant HRE elements (25-27) in RPE cells by CRIPSR/Cas9 to analyze the response in more physiologically relevant context.

Reviewer 2 Report

The manuscript submitted by Klee et al. investigated DEPP expression pattern in the mouse and human retinas as well as the regulatory element under hypoxia. The authors identified three hypoxic response elements (HREs), approximately 3.5 kb from the transcription initiation site, which is responsible for the hypoxic induction of DEPP in the human RPE cells. Interestingly, they also show one of the three HREs is highly conserved in the genomic region across different species, which may have an important physiological function in term of response to hypoxia. Overall, this is a well-written manuscript, a good extension of the author's previous research. The study design is very straightforward and the conclusions are strongly supported by experimental data. However, there are a few issues that need to be addressed to improve research.

Materials and Methods:

There are too many tables, which is not critical to the reader. I suggest that some tables should be moved to supplementary information, such as Table 3-10. In Method 2.3, the RPE / choroid is called "EC". However, it seems that other tissues such as scleral fibroblasts and connected tissues are also included in the "EC". In Method 2.6, the detail of siCTRL sequence is not shown in Table 2. There is no detailed information on how the authors performed statistical analysis in each experiment. Using software like SPSS?

Results presentation

ChIP experiments should be performed to confirm the binding of HIF1A and HIF1B to the identified DEPP HREs In this study, the authors sought to define the expression pattern of DEPP in mouse and human retina and RPE cells. This is a bit exaggerated because the authors only demonstrated the expression of the DEPP gene the retina and RPE/choroidal tissue, not the details of the retinal cells. Immunostaining should be performed to show overall DEPP protein expression in human and mouse retina tissue. Similarly, no protein level has been shown to demonstrate that hypoxia increases DPP expression in human retinas or RPE cells.

Data interpretation

I am curious why the expression of DPP is different in human RPE samples collected from the central and surrounding areas? Is chronic hypoxia different in the central and surrounding areas? Since this is the only evidence that the authors have linked their research to potential human pathologies, this should be discussed further.
